# Development and Verification of a Poultry Management Tool to Quantify *Salmonella* from Live to Final Product Utilizing RT-PCR

**DOI:** 10.3390/foods12020419

**Published:** 2023-01-16

**Authors:** Savannah F. Applegate, April K. Englishbey, Tyler P. Stephens, Marcos X. Sanchez-Plata

**Affiliations:** 1International Center for Food Industry Excellence, Department of Animal and Food Sciences, Texas Tech University, Lubbock, TX 79409, USA; 2Qualicon Diagnostics Division, Hygiena, LLC, 2 Boulden Circle, New Castle, DE 19720, USA

**Keywords:** *Salmonella* enumeration, SalQuant, development, poultry matrices

## Abstract

The United States Department of Agriculture Food Safety and Inspection Service (USDA FSIS) does not maintain a zero-tolerance policy for *Salmonella* in poultry and poultry products, despite being a known food safety hazard throughout the poultry industry. In 2016, USDA FSIS established performance standards for a 52-week moving window with the maximum acceptable percent positive for comminuted turkey (325 g sample) at 13.5% (7 of 52 samples). Based upon FSIS verification sampling results from one 52-week moving window, the *Salmonella* prevalence for each poultry establishment in category 1 (below limit), 2 (meeting limit), or 3 (exceeding limit) are published for public viewing. Moreover, many poultry producers continue to have post-intervention samples test positive. Therefore, the use of quantification would be more valuable to determine the efficacy of process control interventions, corrective actions, and final product Log CFU/g of *Salmonella* to make rapid, within shift, food safety decisions. Therefore, the objectives of these studies are to develop, verify, and validate a rapid and reliable quantification method utilizing RT-PCR to enumerate *Salmonella* in the poultry industry from flock to final product and to utilize the method in an application study. BAX^®^ System SalQuant^®^ is an application of the BAX^®^ System Real-Time PCR Assay for *Salmonella* to enumerate low levels of *Salmonella* with shortened enrichment times. Curve development encompassed inoculating poultry matrix samples at four levels with an ATCC strain of *Salmonella*, with three biological replicates per inoculation level, and five technical replicates being run on the BAX^®^ System for various timepoints, gathering the data, and creating a linear-fit equation. A linear-fit equation was provided for each timepoint. The ideal timepoint, based on the statistical parameters surrounding the equation (R^2^ > 0.80, Log RMSE < 0.60, and enumerable range 0.00 to 4.00 Log CFU/mL (g)) that most accurately estimate *Salmonella* compared to most probable number (MPN), was chosen to be utilized for further studies.

## 1. Introduction

Poultry represents approximately 37% of global meat production, in which the United States (US) produces 48 billion pounds of poultry products annually, making the US the largest poultry producer in the world [1,2]. Compared to other proteins, such as beef and pork, poultry provides necessary nutrients, especially for children, at an affordable price per pound. In 2017, poultry consumption in the US surpassed red meat consumption [3]; therefore, the need to ensure food safety is crucial as consumer market trends promote wholesome and affordable options for the daily consumption of poultry.

As poultry consumption continues to increase, the responsibility for proper cooking, preparation, and handling falls to consumers and retailers. Consumer behavior reports indicate that the majority of consumers do not know how to properly handle raw meat or use a meat thermometer [1,2]. Therefore, foodborne pathogens with high risk of cross-contamination, such as *Salmonella* spp., cause a large economic burden and public health concern worldwide [3,4]. Salmonellosis, the disease caused by *Salmonella* spp., causes over 1.2 million illnesses, 23,000 hospitalizations, and 450 deaths in the US each year; whereas nontyphoidal *Salmonella* causes 11% of all foodborne illnesses, and is the leading cause of hospitalizations (35%) and the leading cause of death (28%) [5]. The Centers for Disease Control and Prevention (CDC) estimates that many of these illnesses are foodborne [3] with over 350,000 illnesses associated with meat, poultry, and egg consumption [6], with two of the top illness-leading serotypes being *Salmonella* Typhimurium and *Salmonella* Enteritidis. Despite *Salmonella* being a known food safety hazard throughout the poultry industry, the United States Department of Agriculture Food Safety Inspection Service (USDA FSIS) does not maintain a zero-tolerance policy for *Salmonella* in raw poultry products. With rates as high as 37.5% in some facilities, a zero-tolerance or adulterant status for *Salmonella* is not a standard that can be met with current process controls and tools available to the poultry industry [7].

*Salmonella* naturally resides in the gastrointestinal tract of avian species; the pathogen often contaminates the carcass during the harvest process through fecal matter [8,9]. Samples such as boot swabs, litter samples, and water samples can give a good indication of the incoming pathogen load on each flock [10,11,12]. Maintaining and eliminating pathogens throughout the poultry processing environment can be performed through chemical and physical interventions to reduce the levels and presence of pathogens such as *Salmonella* [13]. Chemicals such as peroxyacetic acid (PAA), chlorine, bromine, and hot- or cold-water washes are used on poultry carcasses and final products to disrupt the bacterial cell wall, making it difficult for bacteria to survive and reproduce [5]. Additionally, physical interventions are needed to remove the additional contamination from the bird by using brushes and fingers for feather, viscera, and skin removal [14]. In combination, both chemical and physical interventions can be utilized as multi-hurdle interventions to avoid the creation of chemical-resistant bacteria [15].

Chemical, physical, and biological interventions have been developed and validated to combat certain levels of bacteria; however, continued verifications through enumeration are not commonly practiced due to the labor intensiveness and cost of the most probable number (MPN) method, which is the most common enumerative method used currently. Incoming loads of pathogens need to be known to evaluate if the interventions are properly performing within manufacturer specifications. Throughout the poultry industry, specific sample types have been indicated to provide representative results for flocks, processing carcasses, and final products. For primary production flocks, boot or drag swabs are often utilized to estimate the levels of internal *Salmonella* to provide a correlation with the potential level of pathogens the processing environment might encounter [16,17,18]. As identified by USDA—Animal and Plant Health Inspection Service (APHIS), fabric boot covers should be worn over boots specifically worn in poultry barns and sampling 1000 feet of the barn by walking at a normal pace for primary production sampling; whereas drag swab samples are used to drag the barn floor for 15 min or by dragging one length of the barn [18]. Moreover, if *Salmonella* detected from boot swabs were to also be quantified, many production decisions can be made from numerical data such as slaughter order, vaccine verification, and intervention efficacy to create a data-driven connection between primary production and the processing environment.

As the industry relies on prevalence to make decisions, continued changes and improvements go unnoticed due to the nature of presence and absence results having limited statistical ability to impact regulatory decisions. In 2016, the USDA FSIS developed the Poultry Performance Standards (PPS) to promote industry responsibility for the reduction in *Salmonella*. The standards are categorized with maximum percent positives for *Salmonella* allowed on chicken (9.80%) and turkey carcasses (15.7%), chicken parts (15.4%), and ground chicken (25.0%) and turkey (13.5%) products [19]. These percentages are calculated when the USDA FSIS inspectors collect 52 samples weekly within a 52-week moving window; depending on the prevalence level of the samples collected, the facility is ranked into publicized categories depending on if they are below the limit (category 1), meet the limit (category 2), or exceed the limit (category 3) of the PPS. Knowing the incidence rate of *Salmonella* in the facility is beneficial for pathogen presence; however, prevalence does not indicate numerical differences or trends within the facility [6,20]. There is a need, not only in the poultry industry but across the food industry, for a rapid quantification method for *Salmonella* throughout the harvest and processing chains. In the poultry industry, the rapid quantification of *Salmonella* could determine slaughter order, intervention efficacy, and risk of the final product causing a public health concern. Therefore, the objective of this study was to develop and verify a rapid and reliable *Salmonella* quantification method for poultry boot swabs, processing rinses, and ground product as a pathogen management tool using real-time PCR.

## 2. Materials and Methods

### 2.1. Experimental Design

The experimental design for boot swabs, processing rinses (rinsates), and ground samples included three biological replicates, inoculated at five levels (0.0, 1.0, 2.0, 3.0, and 4.0 Log CFU/mL (g), plus 1 uninoculated negative control), three enrichment timepoints, and five technical replicates (PCR lysates) per biological replicate by inoculation level and timepoint combination (Figure 1).

The cycle threshold (CT) values reported from the BAX^®^ System at each timepoint were analyzed using JMP^®^ (Version 14.3.0. SAS Institute Inc., Cary, NC, USA, 1989–2019) to create a linear fit model with the dependent variable as CT and independent variable as inoculation level. Each linear fit model estimation of pre-enriched Log CFU/mL (g) of *Salmonella* was evaluated using R-Squared (R^2^) and Log Root Mean Squared Error (Log RMSE; CT RMSE/slope) values. Linear fit equations per sample type were verified by comparing BAX System *Salmonella* Quantification (SalQuant^®^, Hygiena, Camarillo, CA, USA) estimations to the modified MPN reference method at each inoculation level.

### 2.2. Sample Collection

#### 2.2.1. Boot Swabs

Boot swabs, pre-moistened with 10 mL of Buffered Peptone Water (BPW; Hygiena, Camarillo, CA, USA), were utilized for sample collection following the USDA–APHIS (United States Department of Agriculture–Animal and Plant Health Inspection Service) guidelines [21] in a large commercial poultry production facility. Briefly described, the facilities were sampled by walking at a normal pace in differing locations for 20 feet of the barn prior to changing boot swabs. After use, boot swabs were placed in a sterile Whirl-Pak (Nasco, Fort Atkinson, WI, USA) bag, stored on ice, and shipped overnight to the International Center for Food Industry Excellence (ICFIE) laboratories at Texas Tech University in Lubbock, Texas, and processed within 24 h.

#### 2.2.2. Processing Rinses

Poultry rinses were collected following the USDA MLG (United States Department of Agriculture–Microbiology Laboratory Guidebook) 4.10 guidelines; briefly described by placing a post-chill chicken carcass into a Whirl-Pak poultry rinse bag (Nasco, Fort Atkinson, WI, USA) with 400 mL of neutralizing Buffered Peptone Water (nBPW; Edge Biologicals, Memphis, TN, USA), homogenized by hand by shaking the bird back and forth 30 times, and poured back into the nBPW bottle as a collected rinse. Rinses were shipped overnight on ice to the ICFIE laboratory and processed within 24 h.

#### 2.2.3. Ground Chicken and Turkey

The freshly packed, commercially available, final product of ground chicken and turkey were collected from large commercial USDA-inspected facilities and shipped overnight on ice to the ICFIE Laboratory and processed within 24 h.

### 2.3. Inoculation Procedures

*Salmonella* enterica subsp. enterica Typhimurium ATCC 14,028 and *Salmonella* enterica subsp. enterica Enteritidis ATCC 13,076 (ground chicken only) were removed from the −80 °C freezer and a 10 µL loop of the frozen isolate was streaked onto Brain Heart Infusion agar (BHI; Difco™; Corpus Christi, TX, USA) and incubated for 24 h at 37 °C. After incubation, one typical colony was chosen and placed into 9 mL of BHI broth (Difco™) in triplicate and incubated at 37 °C for 24 h. One percent of the three overnight cultures were transferred into 3–9 mL of BHI broth and incubated at 37 °C for 18 h.

The incubated cultures were serially diluted, 1 mL into 9 mL of Buffered Peptone Water (BPW; Hygiena: Camarillo, CA, USA), and appropriate countable ranges were plated onto BHI agar (Difco™, Corpus Christi, TX, USA) in duplicate and incubated at 37 °C for 24 h. Typical colonies were counted per plate, averaged, and reported as Log CFU/mL. Following enumeration of the pure culture, appropriate amounts of each dilution were utilized to inoculate each level per sample type.

### 2.4. Sample Processing and Curve Development

#### 2.4.1. Boot Swabs

Boot swabs are known to have naturally-occurring *Salmonella*; therefore, a screening step was utilized to eliminate positive samples. Boot swabs (N = 100) were placed in sterile 24-oz filtered Whirl-Pak (Nasco) bags, combined with 100 mL of BPW (Hygiena), and homogenized by hand for 30 s. Five milliliters of the homogenate were added to a sterile tube and incubated for 18 to 24 h and screened for *Salmonella* utilizing the BAX System Real-Time Assay for *Salmonella*. If the BAX System results were positive, the sample was not used. Fifty milliliters of each negative boot swab sample were combined to create a bulk slurry for BAX System SalQuant^®^ linear fit equation development and to achieve standardized background flora across samples.

Thirty milliliters of the bulk slurry were aseptically transferred into 16 sterile 24-oz filtered Whirl-Pak (Nasco) bags. Three biological replicates were inoculated per level (0.00, 1.00, 2.00, 3.00, and 4.00 Log CFU/mL of slurry) with one non-inoculated sample as negative control. Thirty milliliters of pre-warmed 42 °C BAX MP media including Quant Solution™ (Hygiena; 1 mL of Quant Solution per L of BAX MP) were added to each sample, homogenized by hand for 30 s, and incubated at 42 °C for 8, 10, and 12 h.

At each timepoint, the samples were removed from the incubator and analyzed, in quintuplet, using the BAX System Real-Time Assay for *Salmonella*. Samples were removed and replaced within 5–7 min between timepoints with no adverse cooling effects observed. After the last timepoint, samples continued incubating for the remainder of the 18–24 h for prevalence testing. Cycle threshold values from the BAX System result reports were recorded for each timepoint and utilized for linear curve development.

#### 2.4.2. Processing Rinsates

Five 400 mL post-chill poultry carcass processing rinses were combined and homogenized to create a bulk rinse for background flora distribution. Post-chill rinsates were used due to the low presence of *Salmonella* while still having background flora. Thirty milliliters of the bulk rinse were aseptically transferred to 16 sterile Whirl-Pak (Nasco) bags and inoculated at 0.00–4.00 Log CFU/mL with three biological replicates per inoculation level with one non-inoculated negative control. Thirty milliliters of pre-warmed 42 °C BAX MP (Hygiena) media with Quant Solution (Hygiena; 1 mL of Quant Solution per L of BAX MP) were added to each sample. The samples were homogenized by hand for 30 s and incubated at 42 °C for 4, 6, and 8 h.

At each timepoint, the samples were removed from the incubator and analyzed utilizing the BAX System Real-Time Assay for *Salmonella* in quintuplet and replaced into the incubator within 5–7 min to ensure no cooling effects. After the 8 h timepoint, samples continued incubating for the remainder of the 18–24 h for prevalence testing. Cycle threshold values from the BAX System result reports were recorded for each timepoint and utilized for linear curve development.

#### 2.4.3. Ground Chicken and Turkey

Comminuted chicken (*n* = 32) and comminuted turkey (*n* = 32) were created by weighing 325 g into filtered Whirl-Pak (Nasco) bags and inoculated with five levels of *S.* Enteritidis (0.00, 1.00, 2.00, 3.00, and 4.00 Log CFU/g + 1 negative control sample per dilution level with six biological replicates per level). Of the six biological replicates per level, three were combined with 975 mL of BPW for a 1:4 matrix-to-media ratio set and three were combined with 1625 mL of BPW for a 1:6 matrix-to-media ratio set to create primary enrichment solutions. Primary enrichments were homogenized by hand for 1 min.

Thirty milliliters of the primary enrichment solution were transferred into a 24-oz filtered Whirl-Pak (Nasco) bag and combined with 30 mL of pre-warmed 42 °C BAX MP (Hygiena) media with Quant Solution (Hygiena; 1 mL of Quant Solution per L of BAX MP). The secondary enrichment 60 mL solutions were homogenized and incubated at 42 °C for 6, 8, and 10 h. At each timepoint, the samples were removed from the incubator and analyzed in quintuplet utilizing the BAX System Real-Time Assay for *Salmonella*. Samples were replaced into the incubator within 5–7 min after each timepoint to ensure no cooling effects occurred. Samples continued incubating for the remainder of the 18–24 h for prevalence testing. Primary enrichment samples were also evaluated for *Salmonella* prevalence (<1 CFU/mL) by being placed in the incubator at 35 °C for 18–24 h. Cycle threshold values from the BAX System SalQuant^®^ results were recorded for each timepoint and utilized for linear curve development.

### 2.5. Curve Verification

To verify the candidate method, SalQuant^®^, the industry reference method of MPN (USDA MLG 2.05), was utilized with minor modifications during culture confirmation. For boot swabs, rinsates, and ground final poultry products, a 3 × 3 MPN was performed on the negative control sample, and on one biological replicate from 0.00–2.00 Log CFU/mL (g) samples, and a 3 × 5 MPN was performed on one biological replicate from 3.00–4.00 Log CFU/mL (g), based on the enumerable range of the MPN method. The MPN tubes were incubated at 37 °C for 18–24 h. After incubation, *Salmonella* confirmation for each tube was performed using the BAX System Real-Time Assay for *Salmonella*. BAX System presence/absence results were recorded for each tube of the 5-tube MPN, calculated using the MLG 2.05 MPN tables, and transformed into Log MPN/mL. Verification comparisons between the BAX System SalQuant and MPN were performed utilizing the JMP^®^ (Version 14.3.0. SAS Institute Inc., Cary, NC, USA, 1989–2019) statistical program with a statistical significance of alpha 0.05.

## 3. Results

To provide the poultry industry with a rapid and accurate tool for quantification, the tool must meet robust statistical parameters and be comparable to historical data generated by both regulatory and industry stakeholders. Curve development and verification comparison statistical parameters discussed below confirm that real-time PCR coupled with shortened enrichment can be utilized as a rapid quantification method.

### 3.1. Curve Development

The CT values from each timepoint per matrix were evaluated to develop linear-fit curve equations, R^2^, and Root Mean Square Errors (RMSE). The linear equation was used to estimate pre-enriched *Salmonella* levels to mimic the logarithmic growth of bacteria. The R^2^ evaluates the percentage of dependent variable variation a linear model explains and can be observed when the CT value variation depends on the enrichment time and known bacterial inoculation level [4]. The RMSE evaluates the standard deviation of the data and describes how well the CT values estimate the *Salmonella* concentration [5]. Each equation developed was evaluated based upon the statistical parameters of the R^2^ (>0.80), Log RMSE (<0.60), and enumerable range (0.00–4.00 Log CFU/mL (g)) (Figure 2). Developed linear fit equations that did not meet these statistical parameters caused inaccurate estimations compared to known spike levels. Therefore, the importance of meeting statistical parameters throughout the development and verifications of enrichment protocols and linear fit equations is vital to create a rapid tool for quantification.

Statistical parameters for the pre-enrichment estimation of *Salmonella* utilizing linear fit equations ranged from 0.87 to 0.94 R^2^, 0.35 to 0.43 Log RMSE, and 0.00 to 4.00 Log CFU/mL (g). These parameters qualified specific media combinations, incubation temperatures, and shortened incubation timepoints (6–10 h) to establish boot swab, rinsate, and ground final product enrichment protocols (Table 1).

### 3.2. Curve Verification

Most probable number was developed by the USDA to enumerate low levels of *Salmonella* within a food matrix [21]. As seen from the verification data, the method performs best by enumerating low levels of *Salmonella* and not accurately estimating levels of *Salmonella* 3 Log CFU/mL (g) or higher. When enumerating high levels, the method tends to overestimate *Salmonella*. This trend has been observed not only in poultry matrices, but in other food and water matrices as well [22,23,24]. Russek and Colwell (1983) utilized the maximum-likelihood computation model to generate 500 estimates per dilution (2-fold or 10-fold) and 3 Log values (1.00, 1.25, 1.50, 1.75, 2.00, 3.00 Log10 MPN/mL). They concluded that the results from the MPN were highly variable, with a large 95% CI, going as far as stating the MPN estimations were “imprecise” at higher dilutions. Malorny (2008) reiterated that low levels of *Salmonella* can be enumerated accurately with MPN, and higher levels should be enumerated by selective agars to achieve a more accurate estimation [22].

Verification of the candidate SalQuant method to reference modified MLG MPN showed no statistical differences (*p* < 0.001) at any inoculation level. SalQuant estimations on boot swabs, rinsates, and final ground products were verified against MPN estimations (Table 2).

Naturally occurring *Salmonella* in the boot swabs slurry, rinsates, or ground final products was not evident with SalQuant or MPN. SalQuant and MPN compared with low levels (0.00–2.00 Log CFU/mL (g)) of *Salmonella* on each matrix. However, at the 3.00–4.00 Log CFU/mL (g) range, the MPN overestimates *Salmonella* levels. The ground turkey 1:4 MPN overestimated *Salmonella* by 2 Log MPN/g, at the 2 Log CFU/g level, while oversaturating the MPN method by receiving a 3–3–3 result with an upper CI of infinity. Several inoculation levels did not biologically compare, as MPN was over a half of a Log different. The levels were not statistically different due to the increased 95% CI and error associated with the MPN method (Table 2).

## 4. Discussion

### 4.1. Enumerative Methods

The MPN method is laborious, costly, and time-consuming [22,24]. In the poultry industry, there is a need for rapid, affordable, and simple *Salmonella* enumeration methods that delivers results more quickly. More associated error is incorporated into the method with increased labor and transfer steps.

Over the past decade, indicator organism testing has been used in the meat and poultry industry. Indicator organisms are ideally non-pathogenic organisms that behave like a pathogenic organism. Though indicator organism tests are simple and easy to use, they may not be giving insight into the processing of poultry [25]. Enterobacteriaceae (EB) is the family that contains *Salmonella* and *Escherichia coli* and by enumerating EB, pathogen trends can often be observed. Several studies have shown that EB counts do not correlate as well with *Salmonella* levels as previously expected and do not always indicate rebounds or reduction in the pathogen [26,27,28,29]. Furthermore, the best indicator of a pathogen load in a facility is testing for the pathogen and not relying on indicators.

Currently, to test carcass and ground samples from chicken and turkeys, the USDA PPS relies on *Salmonella* prevalence. Most prevalence methods are affordable, not labor intensive, and easy to perform compared to some enumeration methods. However, when final ground product is released with *Salmonella* presence, the level of the pathogen is unknown. With SalQuant, before releasing the final product to consumers, the level of *Salmonella* will already be determined. If the level *Salmonella* is too high, additional hurdles and interventions can be applied to reduce the loads or send the product to be cooked. Moreover, McKee described the 35.0% reduction in *Salmonella* presence from rehang to post-chill birds as seen from the USDA [30]. Though a reduction in *Salmonella* presence is noticeably large, the levels of the pathogen are still unknown. The birds may have had low level contamination at rehang, reduced presence to post-chill, but had a few birds with high *Salmonella* levels causing public health concern. Prevalence and quantification trends for *Salmonella* were observed in De Villena’s (2022) study; however, the reduction magnitude was not seen in quantification (2.27 Log CFU/Sample to 1.94 Log CFU/Sample) as it was in prevalence (>90% to ~40%) from live receiving to rehang samples [29]. Though there was a reduction in both quantification and prevalence, the discrepancies are too variable to be able to utilize prevalence alone as an indication of load.

### 4.2. Primary Production

BAX System SalQuant can act as a tool for on-farm management and estimating *Salmonella* levels within a flock. If a higher level of *Salmonella* is seen in a flock, interventions could be applied, and slaughter order could be determined to reduce the exposure during processing. Several studies have shown that pathogens in boot swab and litter samples are correlated with the level of *Salmonella* within the processing chain and ground product [10,17,31]. Berghaus et al. (2013) identified that *Salmonella* isolation and enumeration utilizing MLG MPN was more sensitive on poultry boot swabs compared to litter samples (*p* = 0.0383). Additionally, Berghaus et al. noticed boot swab *Salmonella* serotypes transferred to the carcass during the harvest process [10]. Santos (2005) conducted a study to enumerate *Salmonella* in boot swabs using MLG MPN and culture confirmation for prevalence then concluded that though the MPN was labor intensive and costly, the benefits of enumeration outweighed prevalence results [31]. The research suggests that implementation of effective interventions to reduce *Salmonella* loads on live birds can reduce the level of pathogens in the final product, ultimately reducing the public health concern for poultry [10,30].

### 4.3. Process Controls

During processing of poultry, various chemical, physical, and biological interventions are combined to reduce the incoming microbial load on each bird. Through effective intervention strategies, most intervention reduce *Salmonella* to safe levels [13]. To visualize these reductions, methods such as Petrifilm™ and MLG MPN provide the poultry industry with a tool to enumerate microorganisms.

Compared to commercially available *Salmonella* enumerative methods, Petrifilm and MPN, SalQuant can accurately estimate the pre-enriched log level of *Salmonella* within poultry rinse samples and provide same-day results compared to 24–48 h. By providing the poultry industry with a tool that can estimate *Salmonella* in processing rinses within 8 h (6 h enrichment and 2 h processing time), the industry can have more clarity in their process. From evaluating intervention efficacy to validating a new process control, SalQuant gives the industry the transparency in their system that is needed to make decisions on how to reduce *Salmonella* levels.

In 2022, De Villena et al. performed a bio-mapping of indicator organisms and pathogens throughout a broiler facility to determine the effectiveness of low and high intervention levels [31]. During their evaluation, De Villena determined that both the aerobic plate count (APC) and EB counts trended with *Salmonella* throughout the processing chain; however, changes in *Salmonella* loads were not observed in the indicator counts [29]. The utilization of SalQuant was essential, in this study, to observe the changes in *Salmonella* throughout the facility with varying levels of chemical interventions [29].

In 2013, Berghaus et al. enumerated *Salmonella* within poultry carcass rinses (pre-harvest, rehang, pre-chill, and post-chill) utilizing MPN. From pre-harvest to post-chill, the prevalence of *Salmonella* decreased from 45.9% to 2.40%. Though the *Salmonella* incidence rate decreased to 2.40%, a reduction of 1.12 Log MPN was achieved through interventions, while still containing 2.32 Log MPN in the eight post-chill carcasses [10]. Additionally, if there is a spike in *Salmonella* in a facility as such in the Berghaus et al. study (2013), control measures can be deployed in time, if SalQuant is used as the quantification method. With prevalence data, positivity can be seen at various locations within the processing chain; however, enumeration through SalQuant can indicate increases and reductions in *Salmonella* while evaluating intervention efficacy at each location. Additionally, the Berghaus et al. study reiterates the need for a rapid and reliable quantification method for same day results for poultry rinses [10].

### 4.4. Final Product and Consumer Risk

Developing an application that can estimate *Salmonella* in ground final poultry products in 10 h (8 h enrichment and 2 h processing time) solves several concerns for the industry. The low level of detection on the comminuted matrices allows for the methodology to excel above competing methods and the potential to quantify *Salmonella* and report prevalence results within the same sample.

Comminuted chicken and turkey equations were built on two dilution factors based on poultry industry need (1:4) and USDA FSIS sampling recommendations (1:6). Depending on the ground sampling method preferences, SalQuant provides an effective and accurate estimation tool for *Salmonella*. In just 10 h, the low level of detection on the comminuted matrices allows for the methodology to excel above competing methods and the potential to quantify *Salmonella* and report prevalence results within the same sample.

Rimet and coworkers (2019) inoculated chickens and turkeys with bioluminescent tagged *S*. Typhimurium and *S*. Heidelberg and tracked the flocks up to 42 days to identify the pathogen in the muscle, skin, and bone. On day 42, no bone or muscle samples tested positive; however, 30.0% of the skin samples were positive for *S*. Heidelberg, concluding that ground samples should have skin removed prior to grinding the meat [12]. Additionally, Peng et al. (2016) found that turkey wing skin had a significantly higher prevalence rate compared to drumstick and thigh skin [32]. These findings are concerning as chicken and turkey skin is commonly used as the fat source for ground poultry products [12,32].

This methodology will allow for a decreased hold time of comminuted products, increasing the amount of product being moved through poultry facilities, and increasing the shelf life associated with the product. From the USDA-FSIS guidelines, raw ground poultry should be stored at refrigeration temperatures (<4 °C) for 1–2 d prior to cooking or freezing [33]. After the recommended time limit, attributes such as color, odor, and taste of the product start to diminish [33,34]. Ground products are released from the facility, and until the pathogen tests come back, the product continues to decrease in shelf life. Furthermore, if a facility could wait for 10 h instead of 1–2 d for pathogen testing results, the product could be released earlier, increasing the shelf life of the product. This could also increase the time in a consumer refrigerator prior to cooking or freezing, while providing a safer final product to consumers.

## 5. Conclusions

The results of these studies demonstrate the ability of BAX^®^ System SalQuant^®^ to adequately enumerate 1–10,000 CFU/mL (g) of *Salmonella* in various farm to final product matrices. SalQuant accurately estimated all levels of *Salmonella* compared to MPN with BAX confirmation that overestimated the higher levels. Compared to MPN, SalQuant is a simple enumeration method that can estimate *Salmonella* within 8–12 h, whereas MPN takes 1 day for BAX confirmation and up to 7 days for culture confirmation.

By providing the poultry industry with a rapid, accurate, and reliable enumeration method for *Salmonella*, facilities can establish slaughter order from on-farm testing, evaluate interventions on farm and in product, while releasing a safer final product to consumers. As determined by De Villena et al. in 2022, reduction in *Salmonella* loads were determined using SalQuant to evaluate low and high levels of chemical interventions in a commercial broiler facility. With the utilization of SalQuant, changes in *Salmonella* within the facility were identified and action was taken, if needed, within the same shift of production. This quick turnaround time to improve food safety will ultimately create a safer, more wholesome product for consumers and decrease the public health concern associated with poultry products.

## Figures and Tables

**Figure 1 foods-12-00419-f001:**
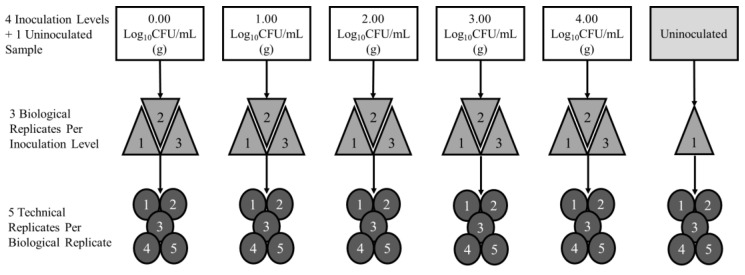
Curve Development Experimental Design for Boot Swabs, Processing Rinsates, Ground Chicken, and Ground Turkey.

**Figure 2 foods-12-00419-f002:**
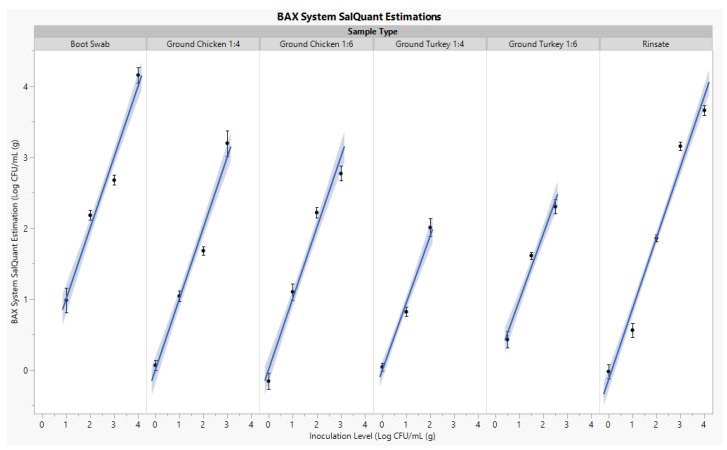
BAX^®^ System SalQuant^®^ Linear-Fit Curve Estimations of Various Poultry Matrices by Inoculation Level.

**Table 1 foods-12-00419-t001:** Curve Development Parameters for Boot Swab, Processing Rinsates, Ground Chicken, and Ground Turkey Matrices.

Matrix	Timepoint (h)	R^2^	Log RMSE *	Enumerable Range ^s^
Boot Swab	10	0.87	0.43	1.00–4.00 Log CFU/mL
Processing Rinsate	6	0.90	0.35	0.00–4.00 Log CFU/mL
Ground Chicken 1:4	8	0.87	0.43	0.00–3.00 Log CFU/g
Ground Chicken 1:6	8	0.87	0.42	0.00–3.00 Log CFU/g
Ground Turkey 1:4	8	0.94	0.19	0.00–2.00 Log CFU/g
Ground Turkey 1:6	8	0.89	0.31	0.00–3.00 Log CFU/g

* Log_10_ Root Mean Square Error (CT RMSE/slope). ^s^ Range in which SalQuant^®^ provided accurate estimation for each matrix for the timepoint.

**Table 2 foods-12-00419-t002:** SalQuant^®^ Curve Estimations Compared to MPN Estimations.

Matrix	Inoculation Level	SalQuant™ Estimation *	MPN ^s^ Estimation *	*p* Value ^γ^
Boot Swab	1.00	1.27	0.56	>0.05
	2.00	2.17	2.15	
	3.00	2.73	4.04	
	4.00	4.17	4.04	
Processing Rinsate	1.00	1.33	1.27	>0.05
	2.00	2.97	2.27	
	3.00	4.25	3.34	
Ground Chicken 1:4	0.00	−0.83	−1.52	>0.05
	1.00	0.67	0.79	
	2.00	1.65	1.32	
	3.00	3.50	2.66	
Ground Chicken 1:6	0.00	−0.16	−0.57	>0.05
	1.00	1.10	1.04	
	2.00	2.23	1.63	
	3.00	2.77	1.79	
Ground Turkey 1:4	0.00	0.11	0.36	>0.05
	1.00	0.83	0.96	
	2.00	2.01	4.04	
Ground Turkey 1:6	0.50	0.55	−0.04	>0.05
	1.50	1.68	1.57	
	2.50	2.33	2.49	

* Reported as Log10CFU/mL (g) or MPN/mL (g). ^s^ Most Probable Number (MPN). ^γ^
*p* Value comparison of SalQuant^®^ to MPN estimation.

## Data Availability

Data available on request from the corresponding author. The data are not publicly available due to privacy from the poultry processing partner that allowed the project to be conducted within their poultry processing facility.

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
