# Peer review of "Development and Verification of a Poultry Management Tool to Quantify Salmonella from Live to Final Product Utilizing RT-PCR"

_foods, 2023, doi:10.3390/foods12020419_

Round 1
Reviewer 1 Report
Review report for foods-2068315
This article discusses the development of a rapid, accurate, and reliable RT-PCR-based enumeration method for Salmonella for use in the poultry industry. It is essential to detect Salmonella in the final product because the USDA does not have a zero-tolerance policy for poultry and poultry products. Thus, the subject of the paper falls within the scope of the Journal, and the scientific content could be interesting. The manuscript is well-organized and written. On the other hand, the manuscript has some punctuation/grammatical errors that need to revise:
In Section “2.3. Inoculation Procedures”, Please write as “Salmonella Enteritidis”. Please consider that when you write it for the first time, use the full form.
In Section “2.4.2. Ground Chicken and Turkey” write it as “ S. Enteridis” by italicizing “S.”
Please add a hyphen for “Real-Time”, “data-driven”, “matrix-to-media”
-In Figure 2, results can not be readable. Values on the curves overlap. The units are also missing for inoculation level and estimation.
-For Table 2, standard deviations for estimations at inoculation levels could be added.
Line 74: Please put a comma before saying “, which” and consider the same situations throughout the manuscript.
Line 234: Please add “a” with statistical significance”
Line 274: Please revise as “highly variable, with a large”
Line 280: Please revise as “final ground products”
Line 307: Please put a comma here “Currently,”
Line 311,319,335 and so on: Please put an article where necessary; “releasing the final product”, “determine the relative health”;” the final product”
Line 383: Please revise as “for a decreased hold time”
Line 390: Please revise as “wait for 10 h”
Author Response
Dear Reviewer 1,
I appreciate you taking the time out of your busy schedule during the holiday season to review this manuscript. Please view the responses the your comments in the file attached.
Thank you and have a wonderful New Year!
Savannah Applegate

Reviewer 2 Report
The subject is interesting and the results are presented in scientific way.
1. What is its novality, when you compare this study with the literature?
2. Further English polishing is needed by a mother-tongue expert.
3. The manuscript needs more updated references to extended the descriptions and supported the results. The manuscript has several typing and grammatical errors and requires improving English.
4. All discussion parts of adsorption experiments results require more comments and references to support the discussion.
5. The conclusion part must be extended with comparison studies with the literature search.
Author Response
Dear Reviewer 2,
I appreciate you taking the time out of your busy schedule during the holiday season to review this manuscript. Below are the responses to each of your comments.
Thank you and have a wonderful New Year!
Savannah Applegate
- What is its novality when you compare this study with the literature?
- BAX System SalQuant was the first commercial RT-PCR platform for quantitative analysis of Salmonella, therefore, unlike any other rapid methods on the market. We feel as the food industry is in dire need of this methodology. Other food safety companies have also understood the food industries need for rapid quantitation as there are a few other options for RT-PCR Salmonella enumeration currently on the market.
- Further English polishing is needed by a mother-tongue expert.
- Three of the authors are native English speakers and one is fluent as English is their second language. We will review manuscript for further grammar and English changes.
- The manuscript needs more updated references to extended the descriptions and supported the results. The manuscript has several typing and grammatical errors and requires improving English.
- Three of the authors are native English speakers and one is fluent as English is their second language. We will review manuscript for further grammar and English changes.
- All discussion parts of adsorption experiments results require more comments and references to support the discussion.
- Additional comments and references to support this discussion were added.
- The conclusion part must be extended with comparison studies with the literature search.
- Additional comments and references to support this conclusion were added.
Reviewer 3 Report
The manuscript examined the develop and verify a rapid and reliable Salmonella quantification method for poultry boot swabs, processing rinses, and ground product as a pathogen management tool using real-time PCR. The design of this paper is well and it may provide some important concern of develop, verify, and validate a rapid and reliable quantification method utilizing RT-PCR to enumerate Salmonella in the poultry industry. All the sentences of this paper were well but it would be better if there were specific results for Salmonella spp. which related public health problems like S. Typhimurium and S. Enteritidis.
Author Response
Dear Reviewer 3,
I appreciate you taking the time out of your busy schedule during the holiday season to review this manuscript. Below are the responses to each of your comments.
Thank you and have a wonderful New Year!
Savannah Applegate
The manuscript examined the develop and verify a rapid and reliable Salmonella quantification method for poultry boot swabs, processing rinses, and ground product as a pathogen management tool using real-time PCR. The design of this paper is well and it may provide some important concern of develop, verify, and validate a rapid and reliable quantification method utilizing RT-PCR to enumerate Salmonella in the poultry industry. All the sentences of this paper were well but it would be better if there were specific results for Salmonella spp. which related public health problems like S. Typhimurium and S. Enteritidis.
Thank you for your clear overview of the manuscript. I followed your recommendations and added more statistics for Salmonella, S. Typhimurium, and Enteritidis.
Reviewer 4 Report
The main goal of this manuscript is to develop and verify a rapid and reliable Salmonella quantification method for poultry boot swabs, processing rinses, and ground product as a pathogen management tool using real-time PCR. The results of these studies demonstrate the ability of BAX® System SalQuant™ to adequately enumerate 1 – 10,000 CFU/mL (g) of Salmonella in various farm to final product matrices. SalQuant accurately estimated all levels of Salmonella compared to MPN with BAX confirmation that overestimated the higher levels. Compared to MPN, SalQuant is a simple enumeration method that can estimate Salmonella within 8 – 12 h whereas MPN takes 1 day for BAX confirmation and up to 7 days for culture confirmation. They concluded that this methodology will allow for a decreased hold time of comminuted products, increasing the amount of product being moved through poultry facilities, and increasing the shelf life associated with the product. I think that the manuscript is convenient with the scope of the journal. The paper could provide information of interest in this field .
Author Response
Dear Reviewer 4,
I appreciate you taking the time out of your busy schedule during the holiday season to review this manuscript. Below are the responses to each of your comments.
Thank you and have a wonderful New Year!
Savannah Applegate
The main goal of this manuscript is to develop and verify a rapid and reliable Salmonella quantification method for poultry boot swabs, processing rinses, and ground product as a pathogen management tool using real-time PCR. The results of these studies demonstrate the ability of BAX® System SalQuant™ to adequately enumerate 1 – 10,000 CFU/mL (g) of Salmonella in various farm to final product matrices. SalQuant accurately estimated all levels of Salmonella compared to MPN with BAX confirmation that overestimated the higher levels. Compared to MPN, SalQuant is a simple enumeration method that can estimate Salmonella within 8 – 12 h whereas MPN takes 1 day for BAX confirmation and up to 7 days for culture confirmation. They concluded that this methodology will allow for a decreased hold time of comminuted products, increasing the amount of product being moved through poultry facilities, and increasing the shelf life associated with the product. I think that the manuscript is convenient with the scope of the journal. The paper could provide information of interest in this field.
Thank you for your review of the manuscript. I appreciate your clear overview and hope this manuscript serves the food industry well, as we believe it will.